

# In vitro anticancer activity of methanolic extract of *Granulocystopsis* sp., a microalgae from an oligotrophic oasis in the Chihuahuan desert

Faviola Tavares-Carreón[1], Susana De la Torre-Zavala[1],
Hector Fernando Arocha-Garza[1], Valeria Souza[2], Luis J. Galán-Wong[1]
and Hamlet Avilés-Arnaut[1]

[1] Facultad de Ciencias Biológicas, Instituto de Biotecnología, Universidad Autónoma de Nuevo
León, San Nicolás de los Garza, Nuevo León, Mexico
[2] Departamento de Ecología Evolutiva, Instituto de Ecología, Universidad Nacional Autónoma de
México, Coyoacán, Mexico

Corresponding author
Hamlet Avilés-Arnaut,
hamlet.avilesarn@uanl.edu.mx

## ABSTRACT

With the purpose of discovering new anticancer molecules that might have fewer side effects or reduce resistance to current antitumor drugs, a bioprospecting study of the microalgae of the Cuatro Cienegas Basin (CCB), an oasis in the Chihuahuan desert in Mexico was conducted. A microalgae was identified as *Granulocystopsis* sp. through sequencing the *rbcL* gene and reconstruction of a phylogenetic tree, and its anticancer activities were assessed using various in vitro assays and different cell lines of human cancers, including lung, skin melanoma, colorectal, breast and prostatic cancers, as well as a normal cell line. The values of $IC_{50}$ of the microalgae methanolic extract using the MTT assay were lower than 20 μg/ml, except that in the lung cancer line and the normal cell line. In vitro, the microalgae extract caused the loss of membrane integrity, monitored by the trypan blue exclusion test and exhibited marked inhibition of adhesion and cell proliferation in cancer cell lines, through the evaluation of the clonogenic assay. Also, typical nuclear changes of apoptotic processes were observed under the microscope, using the dual acridine orange/ethidium bromide fluorescent staining. Finally, the microalgae extract increased the activity of caspases 3 and 7 in skin melanoma, colon, breast and prostate cancer cells, in the same way as the apoptotic inductor and powerful antitumoral drug, doxorubicin. This study shows the anticancer activity from *Granulocystopsis* sp., a microalgae isolated from the CCB.

## INTRODUCTION

Cancer is one of the most important causes of death worldwide and is continuously stimulating the search for new bioactive molecules from natural sources. There is an urgent need of new anticancer drugs because tumor cells are developing resistance against currently available drugs, like vinca alkaloids and taxanes (*Singh et al., 2011*) and some

**How to cite this article** Tavares-Carreón F, De la Torre-Zavala S, Arocha-Garza HF, Souza V, Galán-Wong LJ, Avilés-Arnaut H. 2020.
In vitro anticancer activity of methanolic extract of *Granulocystopsis* sp., a microalgae from an oligotrophic oasis in the Chihuahuan desert.
PeerJ 8:e8686 DOI 10.7717/peerj.8686

anticancer drugs have side effects that can threaten life because they do not discriminate normal and tumoral cells.

During the last decade, microalgae have been extensively used as nutritional or pharmaceutical component to humans and animals. They are considered a potentially new and valuable source of biologically active compounds because can be easily cultured, have short generation times and several anticancer compounds from algae are in clinical or preclinical trials (*Varshney & Singh, 2013*).

Microalgae are unicellular, simple, photosynthetic organisms that have colonized every type of ecological niche. Their adaptive diversification to a multitude of habitats and extreme conditions make them good candidates for drug discovery because they have developed defense compounds to resist changes in solar radiation, temperature, pH, salinity, etc. (*Irigoien, Huisman & Harris, 2004*). One of the most extreme habitats in the north of México is the Cuatro Cienegas Basin (CCB), located in the Chihuahuan desert.

CCB has several hydrological systems which have been listed as a Wetland of International Importance within the international Ramsar Convention (*Souza et al., 2012*). This area is famous for its remarkable biodiversity (*Minckley & Cole, 1968*; *Souza et al., 2006*) despite its extremely unbalanced nutrient stoichiometry between nitrogen (N) and phosphorus (P) (*Elser et al., 2005*; *Lopez-Lozano et al., 2012*; *Souza et al., 2008*). These specific conditions created a unique niche that has persisted generating endemic lineages of microbes (*Souza et al., 2008*, *2018*).

Nevertheless, this very well characterized environment is now dry and most of its macrobiota, extinct. Before the collapse of the aquifer, the biotechnological potential of microalgae from Churince was evaluated in the search for new alternatives against cancer. Therefore, the aim of the present study was to explore the anticancer potential of the methanolic extract of *Granulocystopsis* sp., a microalgae isolated from the Churince intermediate Lagoon in CCB. The antitumor activity was evaluated in breast, colorectal, prostate and skin melanoma, through the evaluation of its cytotoxic activity, morphological analysis, cell adhesive properties and apoptosis induction. This study highlights the importance of conservation of this unique oasis, given its enormous biotechnological potential.

# MATERIALS AND METHODS

## Sampling and isolation of microalgae strain Chu2

Microalgae specimen was hand collected at the intermediate Lagoon in the Churince hydrological system (2° 50.830′ N 10° 09.335′ W), located in CCB, Coahuila, México during period between February and July 2016 under SEMARNAT scientific permit No. SGPA/DGVS/03121/15. For isolation of microorganisms, the sample (fresh water) was homogenized in sterile water and aliquots were placed on Petri dishes containing agar based media: BG-11 (17.6 mm $NaNO_3$, 0.23 mm $K_2HPO_4$, 0.3 mm $MgSO_4 \cdot 7H_2O$, 0.24 mm $CaCl_2 \cdot 2H_2O$, 0.031 mm Citric Acid$\cdot H_2O$, 0.021 mm Ferric Ammonium Citrate, 0.0027 mm $Na_2EDTA \cdot 2H_2O$, 0.19 mm $Na_2CO_3$) supplemented with carbenicillin 50 µg/mL.

**Table 1 Primer sequences used in this study.**

| Primer name | Sequence (5′–3′) | Product size (pb) |
| --- | --- | --- |
| RbcL-192-F | GGTACTTGGACAACWGTWTGGAC | 500 |
| RbcL-657-R | GAAACGGTCTCKCCARCGCAT | |
| RbcLZ-F | CAACCAGGTGTTCCASCTGAAG | 1,100–1,200 |
| RbcLZ-R | CTAAAGCTGGCATGTGCCATAC | |

**Table 2 Strains and plasmid used in this study.**

| Strain/plasmid names | Relevant properties | Source or reference |
| --- | --- | --- |
| *Escherichia coli* DH5α | F−φ80lacZM15 *endA recA hsdR*(r−$_\kappa$m−$_\kappa$) *supE thi gyrA relA* Δ(*lacZYA-argF*)U169 | Laboratory stock |
| PCR$^{TM}$4-TOPO | Plasmid used for sequencing. Km$^R$ | Invitrogen |
| pFT4 | PCR$^{TM}$4-TOPO, Chu2_RbcLZ F/R | This study |
| pFT5 | PCR$^{TM}$4-TOPO, Chu2_RbcL192/657 | This study |

Purity of strain was resolved by sequential restrikes onto new agar plates and a pure strain named Chu2 (Churince strain n°2) was inoculated in liquid BG-11 medium for culture maintenance and up-scaled growth. Cultures were kept in a climate chamber at 20 °C in a 16:8 h light:dark cycle, 70% of relative humidity and 100 μmol photons m$^{-2}$ s$^{-1}$.

## Microalgae morphology

The microalgae Chu2 was observed using the light microscope Olympus BX-53 equipped with phase contrast and a Qimaging camera (model Micropublisher 3.3 RTV) and Q-capture pro 7 software. The morphological identification was performed using the keys for the members of the Phylum Chlorophyta (*John & Tsarenko, 2011*).

## Molecular identification of Chu2 microalgae

Genomic DNA was extracted and used to amplify *rbcL* (rubisco gene) (Table 1). The *rbcL* gene was chosen because it is encoded by the chloroplast genome and is considered a housekeeping gene, and therefore conserved and appropiate for family and genus level phylogenetics. PCR reactions were exposed to the following profile: 35 cycles of denaturation (94 °C for 1 min), primer annealing (55 °C for 1 min), and extension (72 °C for 2 min). The PCR products were ligated into pCR$^{TM}$4-TOPO$^®$ (ThermoFisher Scientific, Waltham, MA, USA) to generate plasmids that were sequenced by LANBAMA-IPICYT, Mexico (Table 2).

## Phylogenetic reconstruction

The *rbcL* sequences were assembled using CodonCode Aligner 5.1 software (CodonCode Corporation, Dedham, MA, USA). The resulting contigs were aligned in Bioedit to build a consensus sequence. The resulting sequence was aligned in the NCBI database (http://www.ncbi.nlm.nih.gov/) using the Basic Local Alignment Search Tool (BLAST) in order to identify the closest related sequences at genus-level affiliations to the Chu2 microalgae *rbcL* gene (GenBank MH370163). After BLAST analysis of the sequenced gene,

a data set of 37 *rbcL* genes from the well characterized and validated genus of the Oocystaceae family (*Stenclova et al., 2017*) was used to construct the phylogenetic tree. The tree was rooted with *Chlorella vulgaris* (Chlorellaceae family) *rbcL* gene. The *rbcL* sequence from Chu2 and the data set were aligned with Clustal V (*Higgins, 1994*) and trimmed to 796 pb by MEGA 7: Molecular Evolutionary Genetics Analysis version 7.0 for bigger datasets (*Kumar, Stecher & Tamura, 2016*). The model selection was performed using statistical and evolutionary analysis of multiple sequence alignments TOPALi v2 (*Milne et al., 2009*). To construct the phylogenetic tree from the genus of Oocystaceae family, the Maximum-likelihood (ML) method was used with MEGA software v. 7 (*Kumar, Stecher & Tamura, 2016*) and the Generalized time-reversible GTR+G parameter as an evolutionary model. The nodes reliability was estimated using the ML bootstrap percentages obtained after 1,000 replications (*Felsenstein, 1985*). Bootstrap values for ML in the range from 0.7 to 1 were marked with black rhombus.

## Preparation of microalgae extract

Pure cultures were inoculated in Erlenmeyer flasks with one L of fresh media (BG-11) and incubated at 25 °C, under 16 h day/8 h dark cycle, in a bioclimate chamber for 2–3 weeks. The resulting media was spun down to separate the microalgae biomass from the broth. Biomass was extracted with MeOH 1:1 (m/v) (Sigma–Aldrich, St. Louis, MO, USA) for 5 days. The crude extracts were evaporated under vacuum at 50 °C (Yamato RE801) to remove methanol residues. For the cytotoxicity assays, the dried methanol extract was dissolved in dimethylsulfoxide (DMSO) to obtain a final concentration of 100 mg/mL (stock) and diluted in $1\times$ PBS.

## Cell lines and cell culture

Cell lines were cultured in RPMI or DMEM with FBS (10% v/v). The cell culture was performed in an incubator at 37 °C and 5% $CO_2$ to ensure growth and viability. The tumor (breast (HTB-22), colorectal (HTB-38), skin melanoma (HTB-72) and prostate (HTB-81)) and Vero normal cell (CCL-81), were purchased from the American Type Culture Collection (ATCC).

## Cytotoxicity assay

Cytotoxicity effects were determined by MTT (3-(4, 5-dimethylthiazolyl-2)-2, 5-diphenyltetrazolium bromide) assays, as previously described by *Van Meerloo, Kaspers & Cloos (2011)*. After incubation for 24 h, cells were treated with various concentrations of *Granulocystopsis* sp. extract and incubated for 48 h. An MTT solution (5 mg/mL) was added to each well and further incubated for 4 h at 37 °C. A medium supplemented with DMSO was used as a control. Doxorubicin (10 μg/mL) treated cells and untreated cells were used as positive control and negative control, respectively. $IC_{50}$ were calculated for each cancer cell line using the equations previously reported, plotting a linear regression curve and using the same in succeeding assays (*Eskandani, Hamishehkar & Ezzati Nazhad Dolatabadi, 2014*). Each concentration of the algal extract was independently assayed three times with three technical replicates.

## Trypan blue exclusion test of cell viability

Different cancer cell lines were grown for 24 h. Subsequently, the cells were exposed to the microalgae extract at the concentration corresponding to their $IC_{50}$ and cell viability was evaluated at 12, 24, 36 and 48 h. After 48 h of treatment, the medium was replaced with fresh medium (without extract) and the cells were cultured for an additional 12 h and 24 h. Trypan blue test was used for the viability assay (*Strober, 2015*). Human cancer and normal cell lines were used without treatment, as negative control. Five technical replicates were performed for each of the three independent experiments.

## Clonogenic assay of cell in vitro

Culture dishes were seeded with 100–110 cells and incubated for 24 h in order to perform the clonogenic assay as previously described (*Rafehi et al., 2011*). Subsequently, the cells were exposed to *Granulocystopsis* sp. extract for 48 h. After treatment, a medium without microalgal extract was added, and cells were cultured for 2 weeks. To determine the number of colonies per plate, the cultures were stained and analyzed using ImageJ software (*Collins, 2007*) and progenitor frequencies expressed as the total number of colonies obtained per 100 cells seeded. Three independent experiments were performed with three technical replicates each.

## Cell morphology and adhesion assay in vitro

Cell attachment assay was carried out with some modifications (*Xia, Wang & Kang, 2005*). Briefly, $5 \times 10^5$ cells were treated with *Granulocystopsis* sp. extract for 48 h in a 6-well plate and then were detached and plated back on a new culture plate. After each incubation period of 6–24 h, the cell attachment status and morphology were observed, and photographs were captured by camera infinity 1–2, Luminera. As a control, cells were cultivated in the same plate without the microalgae extract.

## Dual acridine orange/ethidium bromide fluorescent staining

The AO/EB double staining assay was performed as previously described (*Cohen, 1993*). Briefly, melanoma and prostate cancer cells were treated with *Granulocystopsis* sp. extract for 48 h, trypsinized and stained with AO/EB dye. A Nikon TS100 microscope was used to see and examine the cell suspensions at 400× magnifications. Results were expressed as means ± SE for three independent determinations.

## Caspase assay

Cells were seeded, treated with *Granulocystopsis* sp. methanol extract at their respective $IC_{50}$ values, and incubated for 48 h. Caspases activity was then determined using Caspase-3/7 Fluorescence Assay Kit (Cayman cat. No. 10009135) (*Martinotti, Ranzato & Burlando, 2018*) according to the manufacturer's instructions. Three independent experiments were performed with three technical replicates each.

## Statistical data analysis

Data from the clonogenic assay, caspase activity and AO/EB staining, were expressed as the mean ± SEM from three experiments and GraphPad Prism 7 software was used to perform

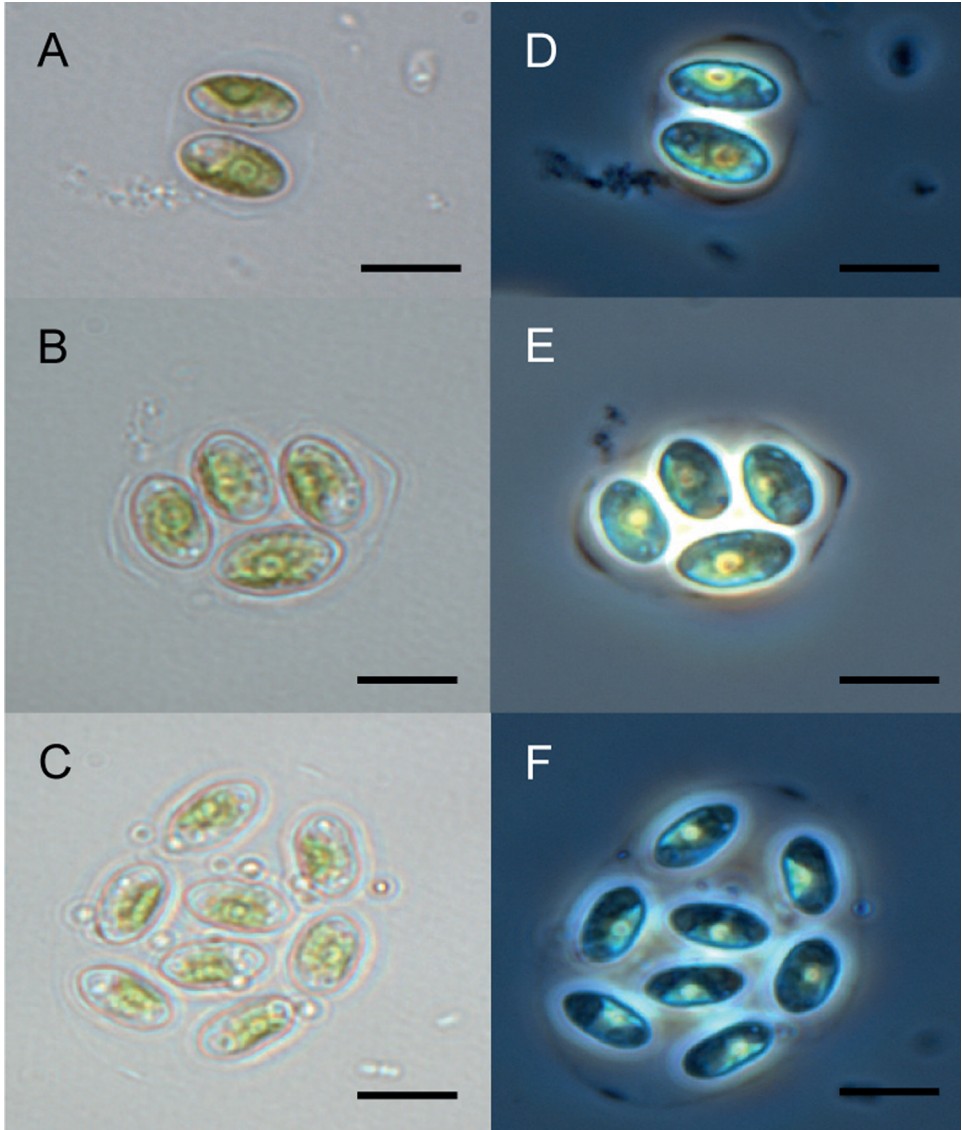

**Figure 1 Microscopy of *Granulocystopsis* sp.** Cells ellipsoidal retained in enlarged parent wall. (A)–(C) Bright field. (D)–(F) Phase contrast. (A) and (D) Colonia with 2 cells. (B) and (D) Colonia with 4 cells. (C) and (F) Colonia with 8 cells. Scale bar 10 μm.               

Students $t$-Test or one-way analysis of variance (ANOVA) followed by Tukey test for multiple comparisons. The significance level was set at $p < 0.05$.

# RESULTS

## Identification of the microalgae strain Chu2

The Chu2 microalgae isolated from in the now extinct Churince hydrological system in CCB, Coahuila, México, was examined by microscopy and it was found to be a Chlorophyta. The cells are ellipsoidal with pointed apices, granular appearance, parietal chloroplast with a pyrenoid, 10–12-micron size, with two cells or multiples of 2 (up to 8) within an expanded lemon-shaped mother cell wall (Fig. 1). Because these characteristics

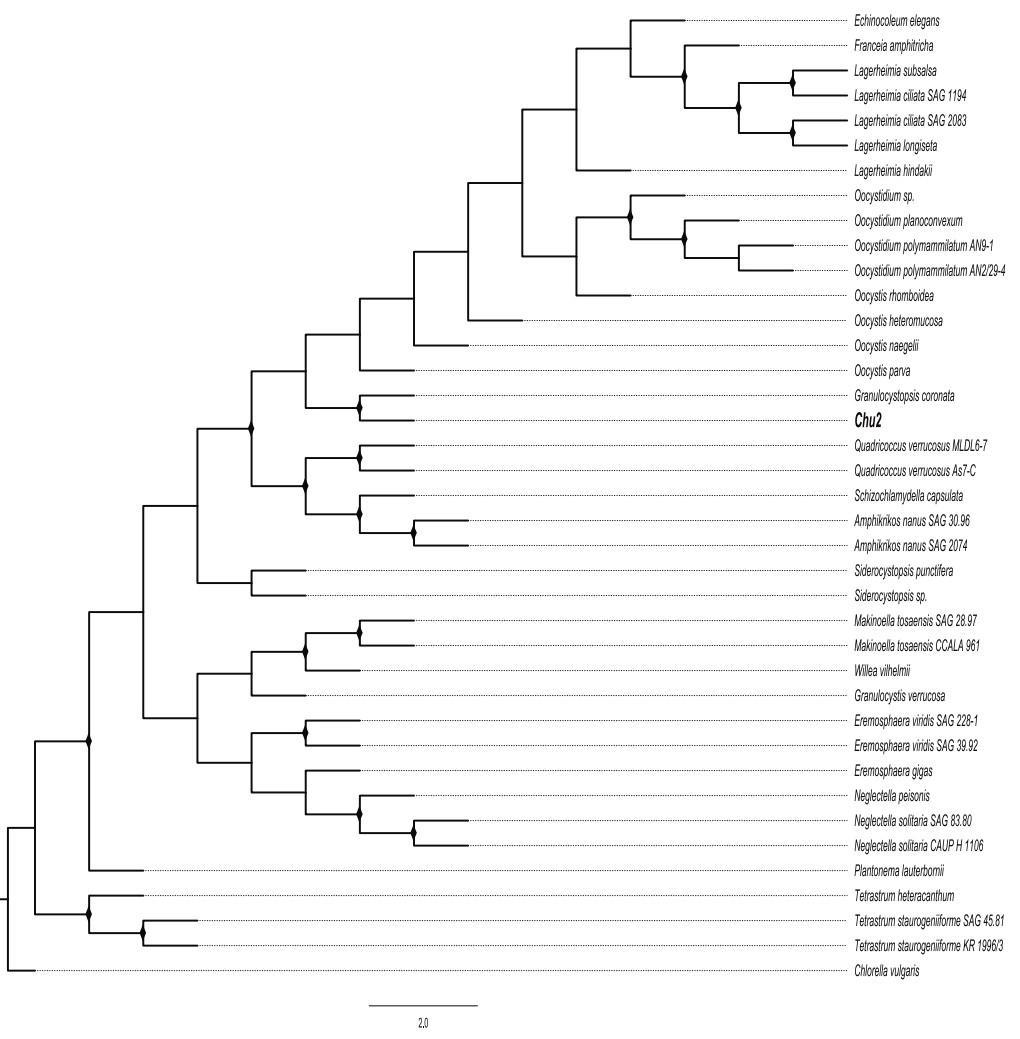

**Figure 2 Phylogenetic tree of Oocystaceae family based on the *rbcL* gene.** Maximum likelihood (ML) method, constructed by the Generalised time-reversible GTR+G parameter as an evolutionary model with 1,000 bootstrap replicates. Bootstrap values for ML in the range from 0.7 to 1 were marked with black rhombus.

are present in some of the members of the Oocystaceae family, the Chu2 *rbcL* gene was amplified with two pairs of primers (Table 1), cloned (Table 2), sequenced and used to construct the phylogenetic tree from the genus of Oocystaceae family in order to identify the closest related homologs in genus-level affiliations to the Chu2 microalgae. Phylogenetic analysis provided the confirmation that the isolate Chu2 belonged to a member of *Granulocystopsis* genus (Fig. 2), and the isolate was designated as *Granulocystopsis* sp. (Chlorellales: Oocystaceae).

## Cytotoxic activity of *Granulocystopsis* sp. extract on different human cancer cell lines

To evaluate the cytotoxic properties of *Granulocystopsis* sp. methanol crude extract, an MTT assay was performed on five human carcinoma cell cultures: lung, prostate, breast,
**Table 3 IC$_{50}$ values (µg/ml) of *Granulocystopsis* sp. methanol crude extract on prostate, breast, colorectal, skin melanoma, and lung cancer cell lines.** Human cancer cell lines were treated with different concentrations of *Granulocystopsis* sp. methanol crude extract in 96-well microculture plates for 48 h. IC$_{50}$ values are expressed as mean ± standard error of mean (S.E.M) of quintuplicate determinations. Different letters represent statistically significant differences determined by one-way ANOVA ($\rho < 0.05$).

| IC$_{50}$ (µg/mL) ± SEM | (µg/mL) ± SEM |
| --- | --- |
| **Cancer cell lines** | |
| Prostate | 13.74 ± 2.06[a] |
| Breast | 16.70 ± 3.09[a] |
| Colorectal | 17.20 ± 2.16[a] |
| Melanona | 17.44 ± 1.64[a] |
| Lung | 1,738.18 ± 1,584.30[b] |
| **Normal cell line** | |
| Vero | 57.02 ± 14.8[b] |

colorectal and skin melanoma. The cytotoxic activity of the microalgae extracts is shown in Table 3. The *Granulocystopsis* sp. extract induced strong cytotoxicity in four cancer cell lines (<20 µg/mL), prostate cancer cells showing striking sensitivity to treatment with the microalgae extract (IC$_{50}$, 13.74 ± 2.06 µg/mL; Table 3). Interestingly, the *Granulocystopsis* sp. extract had no cytotoxic effect on the lung cancer cell line. For that reason, the lung cancer cell line was discarded in the next stage of experiments. The U.S. National Cancer Institute (NCI) has established three groups of crude extracts from natural sources according to their degree of cytotoxicity: inactive (IC$_{50}$ > 100 µg/mL), moderately active (IC$_{50}$ 20–100 µg/mL) and active (IC$_{50}$ < 20 µg/mL) (*Skehan et al., 1990*). The IC$_{50}$ of *Granulocystopsis* sp. microalgae extract on the four cancer lines was less than 20 µg/mL, so the extract is "active" according to the NCI, but also is three times less active in the healthy Vero cell line, showing a slight differential effect between tumor and normal cells.

## Viability (time-dependent) in cells exposed to the extract of *Granulocysptopsis* sp

The trypan blue test was performed to determine changes in the viability of each cell line after being exposed to the *Granulocytostopsis* sp. extract with respect to the time. The assay was performed during 48 h of treatment and 24 h of recovery time after treatment. Interestingly, the greatest decrease in the viability in prostate cells was observed between 0 and 12 h of treatment, between 12 and 24 h of treatment in those of breast cancer and between 24 and 36 h of treatment in those of melanoma and colon. Each cell line responds differently to the extract although the viability of all the cell lines decreased in a time-dependent manner during the treatment with the microalgae extract.

The melanoma, colorectal, and prostate cancer cells showed 70–90% of viability after 24 h of treatment, but breast cells reached only 55% of viability over the same time. After 48 h of treatment, the melanoma, colorectal, and prostate cancer cells showed decreased viability to below 50%, whereas the viability of Vero cells just decreased to 85% (Fig. 3). When 48 h of treatment ended, the cells were incubated with fresh media and monitored for

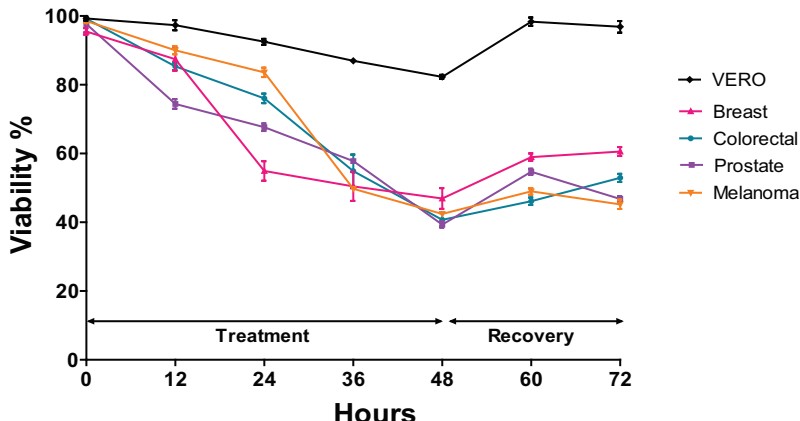

**Figure 3 Changes in cell viability during 48 h of treatment with microalgae extract and 24 h of recovery.** Human cancer cells were treated at the corresponding $IC_{50}$ concentration for each cell line. Cell viability was evaluated by MTT assay. Each data point represents values from three independent experiments ($n = 5$). Error bar indicates mean ± SEM.

24 h. The cancer cells recovered the viability only 10% after 24 h recovery. In contrast, the Vero cell line had almost 100% recovery after the treatment (Fig. 3). Again, the *Granulocystopsis* sp. extract appears to have a cytotoxic and selective effect against prostate, breast, melanoma and colon cancer cells, but with lesser effects on the viability of normal Vero cells.

## Effect of *Granulocystopsis* sp. extract on the proliferation of tumor cell lines

It was investigated whether the microalgae extract could affect the proliferative activity (the ability to form a colony from a single cell), using the clonogenic assay. In the four cancer cell lines treated with microalgal extract, a significant proliferation inhibition was observed (Figs. 4D, 4F, 4H and 4J). The tumor cells treated with the microalgae extract reduced the ability to form colonies by at least 50%, whilst the healthy cell line (Vero) just by 20% (Fig. 4K). According to these results, the *Granulocystopsis* sp. extract has the potential to inhibit the formation of twice tumor colonies in vitro, compared to normal cells.

## Effect of *Granulocystopsis* sp. extract on cell adhesion and morphology of human cancer cells

The effect of *Granulocystopsis* sp. extract on cell adhesion and cell morphology was evaluated by detaching the cells treated with the microalgae extract and plating them in a new plate with fresh medium (extract free). Cells that do not attach to the plate are rounded. Figure 5 shows the level of adhesion and cell morphology between prostate, melanoma, colorectal and breast cancer cell lines with or without the microalgae extract in an interval of 24 h. Vero cells were used as a normal cell. Cells without the extract changed their morphology from round to flattened and adhered to the plate 6 h after incubation (Figs. 5J, 5R and 5HH), reaching almost 100% confluence after 24 h of incubation

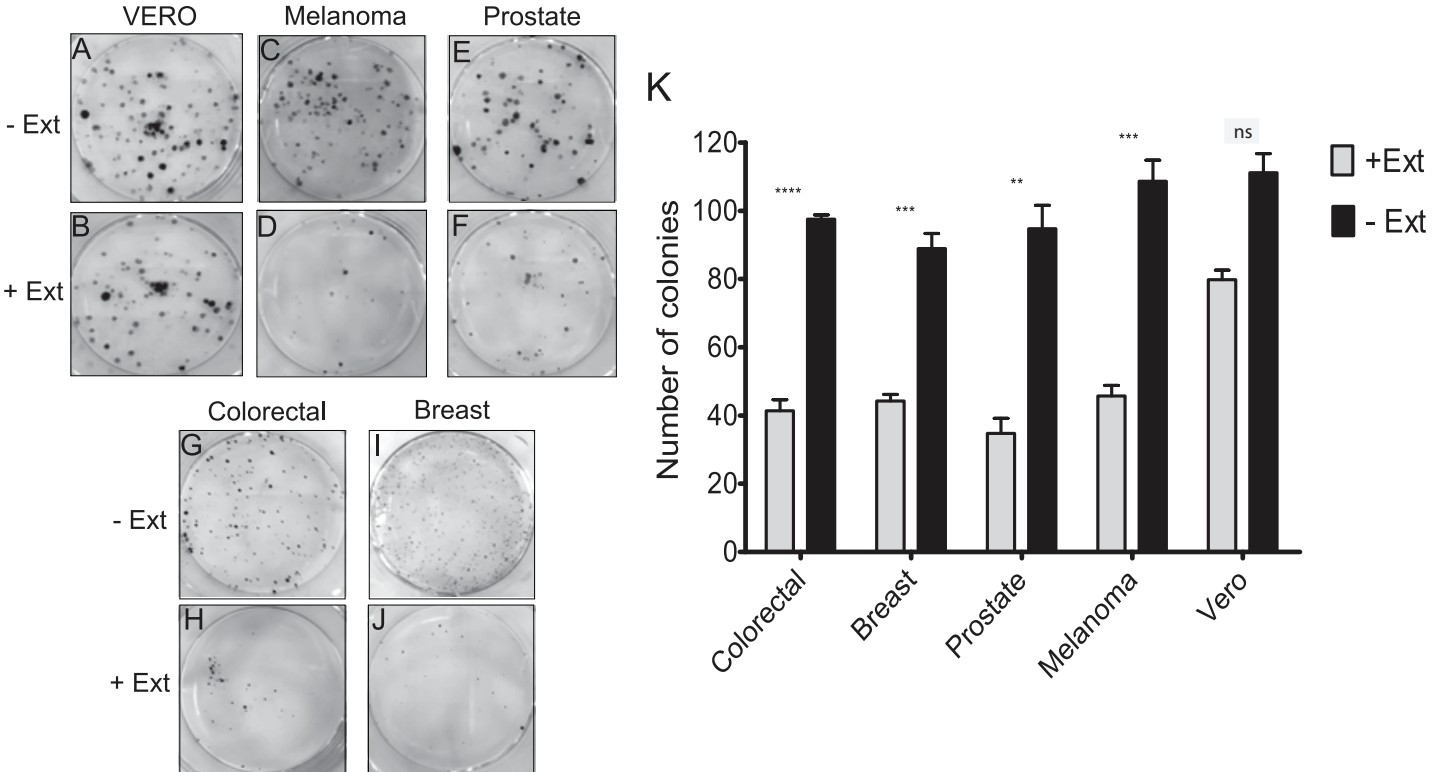

**Figure 4 Colony forming assay of cancer cells in response to treatment with microalgal extract.** Cells of four cancer cell lines were incubated for 10–14 days with microalgal extract at the corresponding $IC_{50}$ concentration. (A, C, E, G and I) Representative images show the clones formed under the control conditions. (B, D, F, H and J) Representative images show the clones formed under the treatment conditions. (K) The number of clones formed after the treatment was counted and presented as histograms. The results are representative of three independent experiments and the level of significance was determined using Student $t$-Test with [ns]representing, $p > 0.05$; [****] represents, $p < 0.0001$; [***] represents, $p < 0.001$ and [**] represents, $p < 0.01$.

(Figs. 5D, 5L, 5T and 5BB). However, the cells treated with *Granulocystopsis* sp. extract, kept their round shape or remained in suspension after 6 h of incubation (especially prostate and breast cells) (Figs. 5N and 5LL), delaying their adhesion to the plaque 12 h. Some treated colorectal, breast and prostate cells (Figs. 5FF, 5NN and 5P) were still unattached 24 h later, hence indicating that the adhesive capability of the treated cells was retarded.

### *Granulocystopsis* sp. extract and apoptosis in human cancer cell lines

To determine whether the cell adhesion, cytotoxic activity and inhibition of cell proliferation by the microalgae extract were due to the induction of apoptosis, the AO/EB staining was assessed to detect nuclear changes and apoptotic body formation. The proapoptotic activity of *Granulocystopsis* sp. extract was investigated with respect to nuclear condensation of cells by fluorescence microscopy. Fluorescence microscopy images clearly showed nuclear changes such as chromatin condensation, nuclear fragmentation and formation of apoptotic bodies in the skin melanoma and prostate cancer cell lines treated with *Granulocystopsis* sp. extract by 48 h (Figs. 6C and 6D). Quantification of the live cells, early and late apoptosis stage and necrotic cell population in

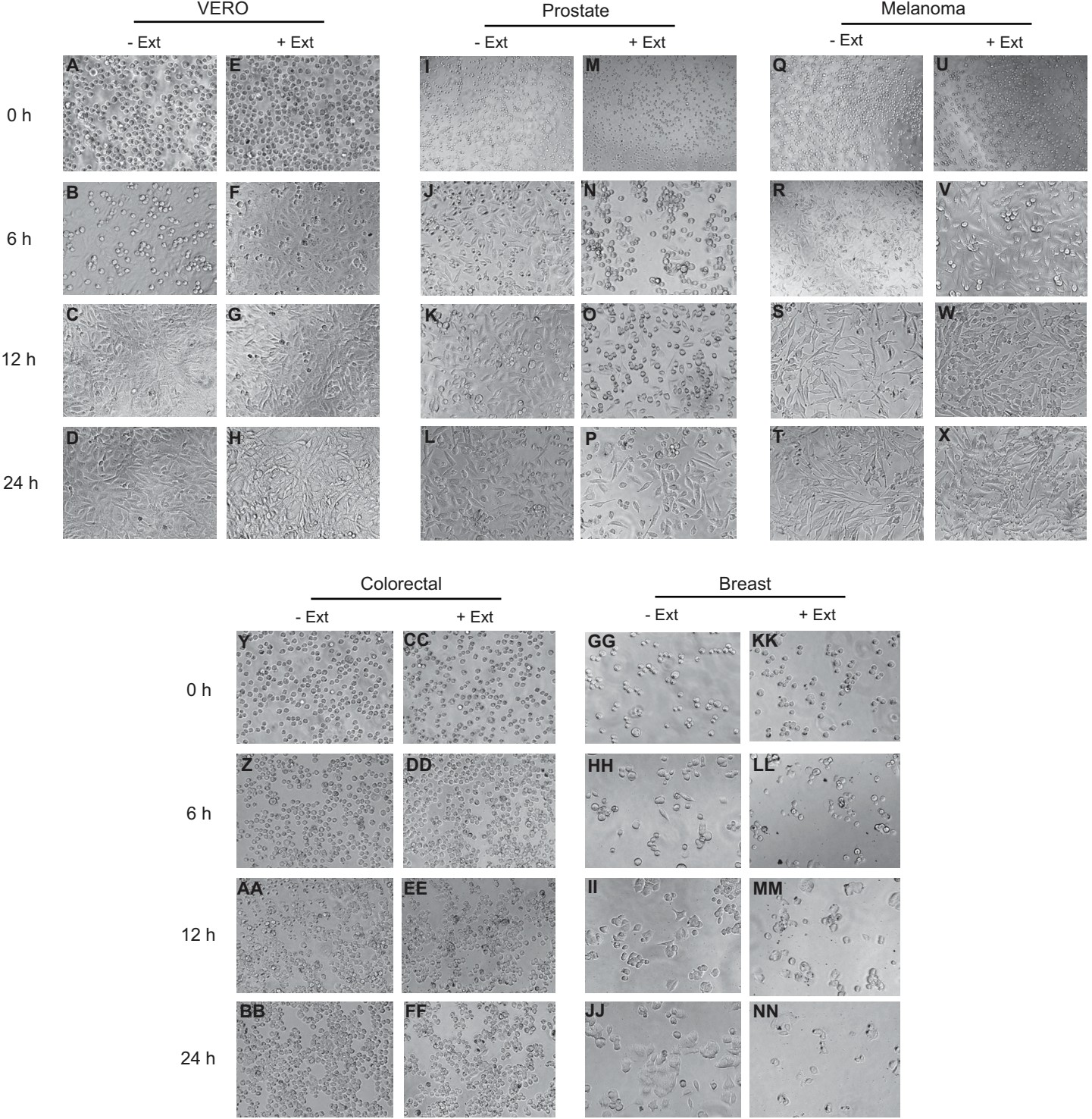

**Figure 5 Effects of microalgae extract on the morphology and cell attachment.** Human cancer cells were treated with IC$_{50}$ corresponding value for each cell line for 48 h and then, the cells were trypsinized and plated on a new culture dish without extract. After a period of 0 h (M, U, CC and KK), 6 h (N, V, DD and LL), 12 h (O, W, EE and MM) and 24 h (P, X, FF and NN), the images were captured with a phase-contrast microscope. Representative results from three independent experiments are shown. +Ext, treated cells. –Ext, control (untreated) cells (I–L, Q–T, Y–BB and GG–JJ). Normal cells (A–H).

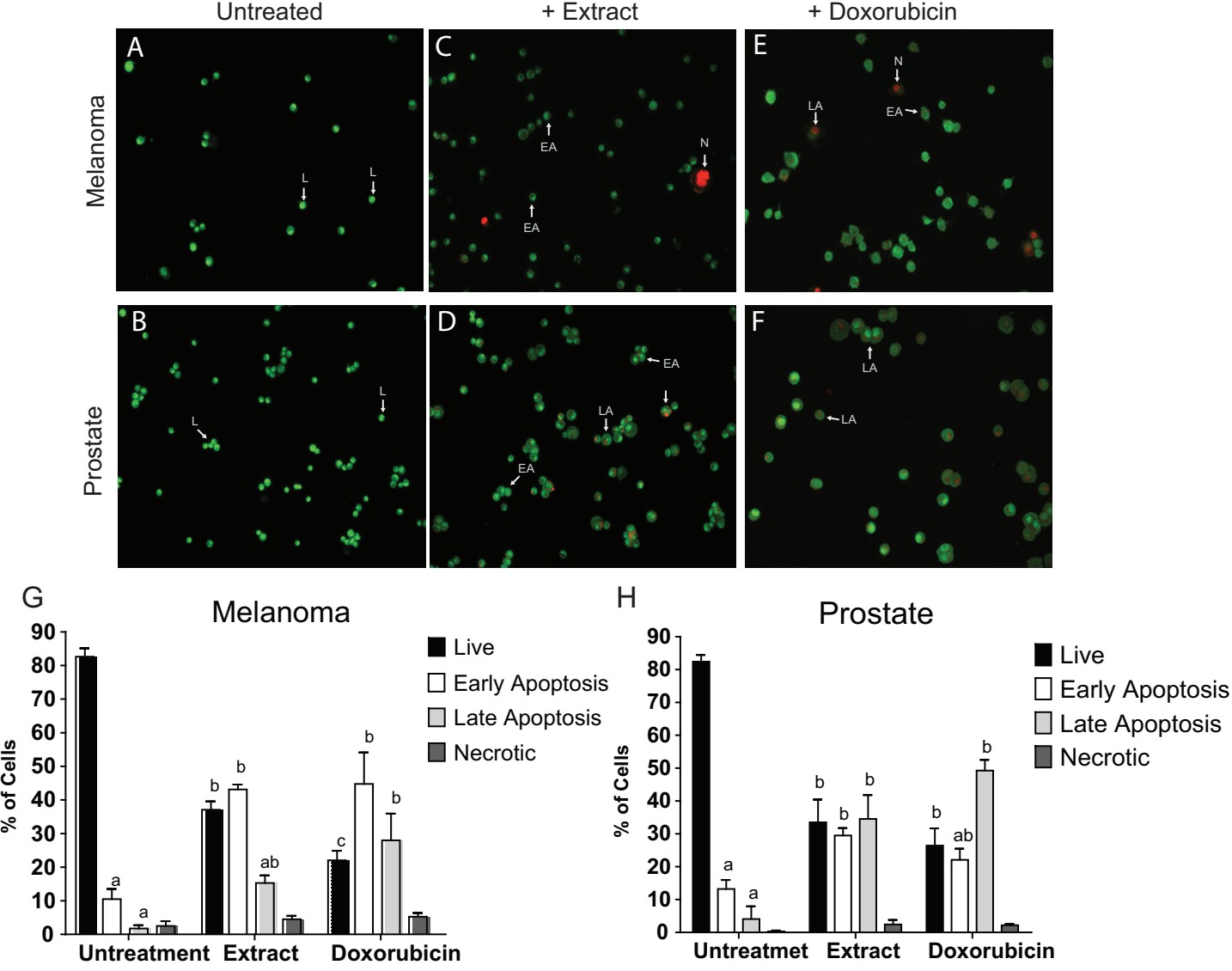

**Figure 6** **AO/EB double stain of human cancer cell lines after a treatment with microalgal extract.** Prostate (D) and melanoma skin (C) cells were treated with microalgal extract at the corresponding $IC_{50}$ concentration. Images represent the control (B and A, untreated cells), treated cells with microalgae extract (D and C) and, cell treated with doxorubicin (10 mg/mL) as a positive control (F and E). Cells were stained with acridine orange and ethidium bromide (AO/EB) after 48 h of treatment. (G and H) Error bar indicates mean ± SEM of three independent experiments. +Ext and +Dox, cells treated with microalgal extract or doxorubicin, respectively. White arrows indicate live (L), early apoptotic (EA), late apoptotic (LA) or necrotic (N) cells. Different letters represent statistically significant differences determined by one way ANOVA ($\rho < 0.05$) between bars with the same color by cell line.

the treated (Figs. 6C and 6D) and control cells (Figs. 6A and 6B) was measured. The skin melanoma and prostate cancer cells increased the early apoptosis stages by 35–45% and the late apoptosis stage by 38–20%, respectively (Figs. 6G and 6H). In addition, the crude extract of the microalgae induced levels of early apoptosis similar to those obtained in cells treated with commercial antitumor compounds, such as Doxorubicin (Figs. 6E and 6H). According to results, it was concluded that the *Granulocystopsis* sp. extract can induce in vitro apoptotic events in skin melanoma and prostate cancer cell lines.

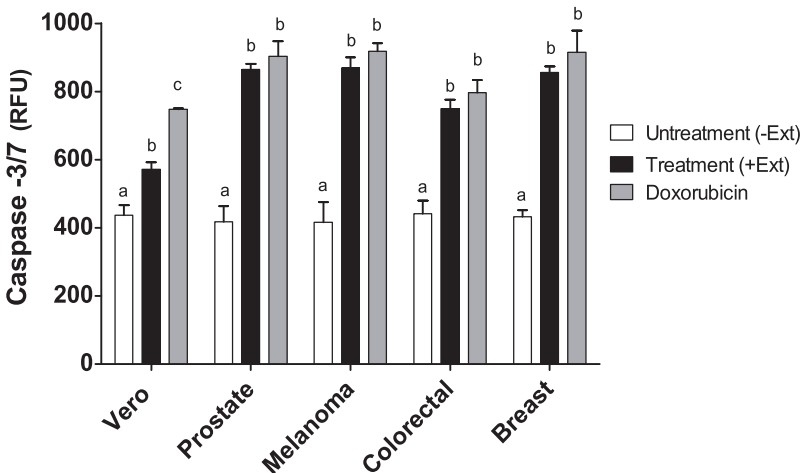

**Figure 7** Caspase-3/7 activty on cancer cell lines treated with *Granulocystopsis* sp. microalgal extract. Quantitative assessment of caspase activity in prostate, melanoma, colorectal and breast cancer cell lines. Vero is a normal cell line. Cells were treated with *Granulocystopsis* sp. extract at the corresponding $IC_{50}$ concentration for each cell line. Error bar indicates the standard error of the mean of Relative Fluorescence Units (RFU) of three independent experiments. Different letter represents statistically significant differences determined by one way ANOVA ($\rho < 0.05$) between bars with different color by cell line.

## Caspase-3 and -7 activities in cancer cell lines treated with *Granulocystopsis* sp. extract

Caspases are members of the aspartate-specific cysteinyl protease family and are involved in the regulation of apoptosis and inflammation (*Kaufmann et al., 1993*). Therefore, to corroborate apoptosis induction by *Granulocystopsis* sp. crude extract on the cancer cell lines, caspase-3 and -7 were measured. Figure 7 shows that the activity of caspases 3 and 7 was increased twice in the tumor cells treated with the *Granulocystopsis* sp. extract, compared to untreated cancer cells. On the other hand, in Vero (normal) cells, the positive control treated with doxorubicin showed a higher activation than Vero cells treated with microalgae extract. No differences in caspase activity were observed between cancer cells treated with doxorubicin and those treated with the microalgae extract. Together, these experiments strongly support the conclusion that *Granulocystopsis* sp. extract has cytotoxic activity induced by apoptotic activation mediated by caspases 3 and/or 7.

## DISCUSSION

In the last three decades, more than 50,000 natural products have been discovered from marine microorganisms, many of them with biomedical applications (*Newman & Cragg, 2012*; *Wiese et al., 2009*). Analysis of molecules produced by aquatic organisms has shown that microalgae synthesize a large number of compounds with different biotechnological applications, including those with anticancer activity. Cyanobacteria, diatoms and chlorophytes are an emerging source for the discovery of new drugs because they are organisms that grow in under-explored extreme environments.

In an attempt to discover new anticancer molecules that may have fewer side effects or reduce resistance to current anticancer drugs, a bioprospecting study of microalgae from CCB, an hyper-diverse oasis in the Chihuahuan desert in Mexico was conducted. A microalgae (strain Chu2) was isolated from the Churince lagoon, and its microscopic morphology coincided with a member of the Oocystaceae family. The molecular identification of the microalgae was carried out using the *rbcL* gene (which encodes RuBisCO, a fundamental enzyme in the process of photosynthesis), according to the recommendation of the Consortium Barcode Of Life for the identification of photosynthetic organisms (*CBOL Plant Working Group, 2009*). The DNA sequence was analyzed using BLAST, showing 100% coverage and percent identity with the *rbcL* gene previously reported for *Granulocystopsis coronata*. This information was confirmed by a phylogenetic analysis with other members of the Oocystaceae family. *Granulocystopsis* is a genus of freshwater microalgae from the Oocystaceae family with 6 names of species taxonomically accepted: *G. calyptrata, G. coronata, G. decorata, G. elegans, G. reticulata* and *G. subcoronata* (*John & Tsarenko, 2011*). However, research papers about this genus are limited to its taxonomy and there are no reports about its biotechnological potential. Although the most abundant photosynthetic aquatic microorganisms reported in CCB are cyanobacteria and diatoms (*Pajares et al., 2012*; *Winsborough, Theriot & Czarnecki, 2009*), the Churince lagoon used to have several green microalgae, an unexplored group of organisms which, like the Chu2 strain (identified as *Granulocystopsis* sp.), are adapted to live in oligotrophic conditions, possibly by modifying their metabolism and generating molecules with possible cytotoxic activity against fast-growing eukaryotic organisms in order to avoid competition and obtain phosphorous and nitrogen from the lysed cells in their surroundings. This selective cytotoxicity may explain why they target the fast-growing cancer cells in skin melanoma, colorectal, breast, and prostate cancer without damaging normal cells.

Interestingly, in the cell lines evaluated, the $IC_{50}$ value obtained was from 13.74 µg/mL to 17.44 µg/mL, whereas normal cells treated with the microalgae extract showed an $IC_{50}$ value of 57.02 µg/mL (three times higher than cancer cells). This result revealed that *Granulocystopsis* sp. extracts have cytotoxic activity which might be helpful in preventing the cancer's progress, especially when it is compared against the activity of other extracts of isolated microalgae from Mexico, such as, *Chlorella sorokiniana* ($IC_{50}$ 460 µg/mL) and *Scenedesmus* sp. ($IC_{50}$ 362 µg/mL) against lymphoma cells (*Reyna-Martinez et al., 2018*), or other microalgal extracts from *Alexandrium minutum* ($IC_{50} > 50$ µg/mL) against melanoma cells (*Lauritano et al., 2016*), *Haematococcus pluvialis* ($IC_{50}$ 27–72 µg/mL) against colon, breast and hepatocellular carcinome (*El-Baz et al., 2018*), *Dunaliella salina* ($IC_{50} > 400$ µg/mL) against neuroblastoma cells (*Atasever-Arslan et al., 2015*), *Scenedesmus obliquus* ($IC_{50}$ 24–93 µg/mL) against colon, hepatocelullar and breast cancer cells (*Marrez et al., 2019*) and *Chloromonas reticulata* ($IC_{50} > 50$ µg/mL) (*Suh et al., 2019*) and *Micractinium* sp. ($IC_{50}$ 100 µg/mL) against colon cancer cells (*Suh et al., 2018*). Additionally, it was corroborated that the microalgae extract has a cytotoxic effect at the level of membrane integrity, using the trypan blue vital dye, which is excluded by an intact cell membrane (*Strober, 2015*). When the cancer cell lines were treated for 2 days in
the presence of microalgae extract, the capability to recover the viability decreased significantly, while the healthy cell line recovered 100% viability 12 h after removal of the extract. These results suggest that the extract of *Granulocystopsis* sp. affects the viability of cancer cells in a time-dependent manner and probably could have tumor-specific activity with minor side effects for normal cells.

The ineffectiveness of currently available treatments is mainly due to the invasive and metastatic properties of malignant cancer cells (*Lee et al., 2011*). Proliferation and cell adhesion are crucial steps that play a significant role in cancer progression and metastasis. The metastatic spread is determined by the cell-cell interactions of cancer cells with endothelium, due to their ability to adhere strongly before they can colonize and establish a secondary tumor in a new place (*Chambers, Groom & MacDonald, 2002*). Data obtained from the clonogenic assay, the adhesion and cell morphology tests, showed that extract of *Granulocystopsis* sp. reduced the ability of cancer cells to form colonies and decreased the attachment ability compared to untreated cells. These results suggest a potential antimetastatic activity of *Granulocystopsis* sp. extract, which could be evaluated through migration and cell invasion assays and elucidate possible action mechanisms where some cytoskeleton components were involved. Apoptosis is characterized by a number of characteristic morphological changes in the structure of the cell, together with a number of enzyme-dependent biochemical processes. The result is the clearance of cells from the body, with minimal damage to surrounding tissues and it is the mechanism facilitating the action of many chemotherapeutic drugs. Failure of apoptosis and the resultant accumulation of damaged cells in the body can lead to malignant transformation and result in various forms of cancer (*D'Arcy, 2019*). One technique used to visualize the early and late stages of apoptosis is AO/EB fluorescent staining (*Ribble et al., 2005*). Our results showed that the microalgae extract activated the apoptosis mechanism in tumor lines. Interestingly, the microalgae extract induced the same level of cells in early and late apoptosis with respect to the anti-cancer compound doxorubicin, suggesting that the extract might contain a more potent compound or a mixture of compounds working in synergy, and therefore, further analyses are required for chromatographic separation and identification of active compounds by NMR, mass spectrometry, etc.

The initiation of apoptosis is dependent on the activation of a series of cysteine-aspartic proteases known as caspases (*Shi, 2002*). Caspases can be divided into caspase-8 and -9 (initiator caspases) and caspase-3 and -7 (executioner caspases). Both initiator caspases can activate the caspase-3 or -7, which are mainly responsible for the final stages of apoptosis, which consist of chromatin segregation, nuclear condensation, and finally DNA fragmentation (*Pojarova et al., 2007*; *Yang et al., 2006*). Our results showed that apoptosis occurred in melanoma, prostate, colorectal and breast cancer cells treated with microalgal extract, activating caspase-3 and -7, which were increased manifold over the basal level of untreated cells. Again, the level of activation of caspases was similar among the cancer cells treated with the extract and the compound doxorubicin, which strengthens our proposal for the extract of *Granulocystopsis* sp. as a good candidate as an anti-cancer drug, which can promote apoptosis in cancer cells via the mitochondrial-dependent intrinsic pathways. The intrinsic pathway can be triggered by irradiation, oxidative stress, hypoxia or

cytotoxic drugs (*Jan & Chaudhry, 2019*). To discover signal transduction involved in triggering apoptosis mediated extract *Granulocystopsis* sp., detection of intracellular reactive oxygen species level, analysis of mitochondrial membrane potential and Western blotting analysis are required to establish the mechanisms of action of the extract and the participation of Bax/Bak (pro-apoptotic protein inserted into mitochondrial membrane), Bcl-2 (inhibits production of cytochrome c), Cytochrome c (released into the cytosol), Caspase-9 (induced by cytochrome c), and other pro-apoptotic proteins from the intrinsic pathway like Smac/Diablo, Apaf-1, among others, leading to the activation of caspase-3. Because there are studies that confirm the participation of polyphenols in the induction of apoptosis in tumor cells (*Sharif et al., 2010*; *Walter et al., 2010*), more experiments are required to demonstrate if any phenolic compound present in the *Granulocystopsis* sp. extract could be initiating the transduction signal from the intrinsic pathway.

Based on our results, the microalgal extract may be useful for the future development of anti-metastatic therapeutic agents. The current research aimed at the description of the molecular mechanisms of the anticancer properties of the microalgae extract, as well as the elucidation of the bioactive molecule, is being performed.

## CONCLUSIONS

The current study represents the first report showing the anticancer activity derived from *Granulocystopsis* sp., an isolated microalgae from the Chihuahuan desert. The microalgae methanolic extract inhibited cell proliferation, showed time-dependent cytotoxic activity, modified morphology, decreased cell adhesion and induced apoptosis by activating caspases-3/7 in breast, colon, prostate and skin melanoma cancer cell lines, but showed less pronounced effects on normal cells.

## ACKNOWLEDGEMENTS

We would like to thank our colleagues Dr. Mariana Elizondo Zertuche and Álvaro Colín Oviedo for their support for photos with the fluorescence microscope at Departamento de Microbiología, Facultad de Medicina, UANL. We thank Alberto Ramos-Silva for technical assistance, Jorge Alberto Balderas-Soriano for the preliminary results and Abigail Mata-Aguilar for keeping the CCB microalgae culture collection. We also want to thank "Centro de Bachillerato Tecnológico Agropecuario #22" for providing facilities during the sampling period and we thank SEMARNAT for access to the CCB Natural Protected Area and permission to sample there.

### Funding

This research project was supported by two grants from The Program for Scientific and Technological Research (PAICyT-UANL No. CN373-15) and Consejo Nacional de Ciencia y Tecnología (CONACyT, No. 239695). The fluorescence microscope came from grant CONACYT-INFRA 2015-251142 at Departamento de Microbiología, Facultad de

Medicina, UANL. Faviola Tavares-Carreón was supported through a Postdoctoral grant (code 173833) and Héctor Fernando Arocha-Garza was supported by a doctoral research scholarship (code 490680) from CONACyT. The funders had no role in study design, data collection and analysis, decision to publish, or preparation of the manuscript.

## Grant Disclosures

The following grant information was disclosed by the authors:
Scientific and Technological Research (PAICyT-UANL): CN373-15.
Consejo Nacional de Ciencia y Tecnología (CONACyT): 239695.
Departamento de Microbiología, Facultad de Medicina, UANL: CONACYT-INFRA 2015-251142.
Postdoctoral Grant: 173833.
Consejo Nacional de Ciencia y Tecnología (CONACyT): 490680.

## Competing Interests

Valeria Souza is an Academic Editor for PeerJ.

## Author Contributions

- Faviola Tavares-Carreón conceived and designed the experiments, performed the experiments, analyzed the data, prepared figures and/or tables, authored or reviewed drafts of the paper, and approved the final draft.
- Susana De la Torre-Zavala analyzed the data, prepared figures and/or tables, authored or reviewed drafts of the paper, did the field work and sampling in CCB, and approved the final draft.
- Hector Fernando Arocha-Garza analyzed the data, prepared figures and/or tables, authored or reviewed drafts of the paper, and approved the final draft.
- Valeria Souza analyzed the data, authored or reviewed drafts of the paper, and approved the final draft.
- Luis J. Galán-Wong analyzed the data, authored or reviewed drafts of the paper, and approved the final draft.
- Hamlet Avilés-Arnaut conceived and designed the experiments, analyzed the data, prepared figures and/or tables, authored or reviewed drafts of the paper, did the field work and sampling in CCB, and approved the final draft.

## Field Study Permissions

The following information was supplied relating to field study approvals (i.e., approving body and any reference numbers):

SEMARNAT provided access to and permission to sample in the CCB Natural Protected Area (scientific permit No. SGPA/DGVS/03121/15).

## DNA Deposition

The following information was supplied regarding the deposition of DNA sequences:

The Chu2 microalgae rbcL gene is available at GenBank: MH370163.

## Data Availability

The raw data are available in the Supplemental Files.

## Supplemental Information

Supplemental information for this article can be found online at http://dx.doi.org/10.7717/peerj.8686#supplemental-information.

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
