# Peer review of "In vitro anticancer activity of methanolic extract of Granulocystopsis sp., a microalgae from an oligotrophic oasis in the Chihuahuan desert"

_PeerJ, doi:10.7717/peerj.8686_

## Round 0.1 · original submission · Minor Revisions

Please address all the critiques of the reviewers and amend your manuscript accordingly.

Reviewer 1 ·

Basic reporting

Structure of the manuscript is acceptable.

Experimental design

There are general screening experiments toplam understand apoptotic effect of the extract.

Validity of the findings

Findings are acceptable for the manuscript.

Additional comments

The manuscript is acceptable but it needs more detail abort signal transduction of the extract. Discussion should be extended.

·

Basic reporting

Clear and unambiguous, professional English used throughout
Literature references, sufficient field background/context provided
Professional article structure, figures, tables.

Experimental design

aim of work is very clear but it can be complete by identification of bioactive compounds that found in the methanolic extract

Validity of the findings

the novelty of this study can be complete by connect the cytotoxicity by compounds that cause that, but the work is very important in isolation, purification and identified Granulocystopsis sp. and by morphological (where is the reference of morphological identification?) and genetic. then extraction bioactive metabolites, then study the effect of this extract on cancer cell lines.

Additional comments

Title: In vitro anticancer activity of methanolic extract from Granulocystopsis sp., a microalgae from an oligotrophic oasis in the Chihuahuan desert
Very expressive
Abstract: Lines 26 – 29: Delete "Many types of biologically active components have been identified in microalgae and used in medicinal applications, but the success of biotechnological potential rises when biological diversity allows the discovery of novel chemical diversity such as the one that can be found in extremophiles or endemic organisms of untapped environments". This is abstract not introduction begin with the objective or the aim of work followed by summarized methodology and results and end by brief conclusion. The abstract should be rewrite.
Key words: very long, can you summarized it"cytotoxic activity=anticancer activity" also if you write microalgae you should not write Granulocystopsis
Introduction: Prepared well, but it is very long
Materials and methods: writing very well but it should summaried
Also, where is the reference and manuals of morphological identification?
Results: the methanolic extract contains mainly ohenolic compounds. why the authors did not identified the compounds in this extracts using HPLC-MS or at least identified the phenolic compounds by HPLC.
Conclusion: very long, summaried it main points that observed the impact of your study and the novelty of your data. The conclusion should be rewrite.

·

Basic reporting

The title should be changed to: In vitro anticancer activity of methanolic extract of
Granulocystopsis sp., a microalgae from an oligotrophic oasis in the Chihuahuan desert.
Generally, it is an important idea and can improve anticancer drugs by using natural resources from algae. but it needs to improve the languages. A lot of verbs in present and it should be in past tense.
Phrases are too long and should be more clear and shorter as in the abstract (Example: First one phrase).
Using the form we is not appropriate and it is better to use the passive tense.
Line 31: delete "we decide" and add "was conducted" in line 32 after the word"Mexico.
lines 32-26: too long phrase
line 40: mention to the methanolic extract
line 40: of instead of from.
41: except that in
41: MTT should be written in whole abbreviation
84-88: rewrite and correct grammatically
91: "methanolic extract instead of" instead of "methanolic extract from"
126: we is not appropriate
225: write the reference of the manufacture instruction

Experimental design

Is well designed but should be in brief not like in thesis.
Reference of morphological identification should be added

Validity of the findings

good but the titles of all finding should not be written as a normal phrases. Titles should not contains verbs.

Additional comments

Check it grammatically well and carefully

---

## Round 0.2 · accepted · Accept

All critiques of all reviewers were adequately addressed and the manuscript was revised accordingly. Therefore, I am glad to accept your manuscript now.